# The Effect of Multi-Walled Carbon Nanotubes on the Heat-Release Properties of Elastic Nanocomposites

Alexander V. Shchegolkov [1], Mourad Nachtane [2,*], Yaroslav M. Stanishevskiy [3], Ekaterina P. Dodina [3], Dovlet T. Rejepov [3] and Alexandre A. Vetcher [3,4,*]

1   Department of Technology and Methods of Nanoproducts Manufacturing, Institute of Technology, Tambov State Technical University, 392000 Tambov, Russia
2   S Vertical Company, 92290 Paris, France
3   Institute of Biochemical Technology and Nanotechnology (IBTN), Peoples' Friendship University of Russia (RUDN), 6 Miklukho-Maklaya St, 117198 Moscow, Russia
4   Complementary and Integrative Health Clinic of Dr. Shishonin, 5 Yasnogorskaya St, 117588 Moscow, Russia
*   Correspondence: mourad.nachtane@svertical.com (M.N.); avetcher@gmail.com (A.A.V.)

**Abstract:** Of great importance in materials science is the design of effective functional materials that can be used in various technological fields. Nanomodified materials, which have fundamentally new properties and provide previously unrealized properties, have acquired particular importance. When creating heating elements and materials for deformation measurement, it is necessary to understand the patterns of heat release under conditions of mechanical deformation of the material, as this expands the potential applications of such materials. A study of elastomers modified with multi-walled carbon nanotubes (MWCNTs) has been carried at the MWCNTs concentration of 1–8 wt.%. The modes of heat release of nanomodified elastomers at a voltage of 50 V at different levels of tension are reported. The increment of the MWCNTs concentration to 7 wt.% leads to an increment in the power of heat emissions. It is worth noting the possibility of using the obtained elastomer samples with MNT as sensitive elements of strain sensors, which will allow obtaining information about physical and chemical parameters following the principles of measuring the change in electrical resistance that occurs during stretching and torsion. The changes in conductivity and heat emission under different conditions have been studied in parallel with Raman mapping and infrared thermography. The reported studies allow to make the next step to develop flexible functional materials for the field of electric heating and deformation measurement based on elastic matrices and nanoscale conductive fillers.

**Keywords:** MWCNT; catalyst; organosilicon compound; stretching; torsion; heating; percolation; modification

## 1. Introduction

In the contemporary development of new types of functional materials, the direction associated with the study of the influence of different types of fillers on the physicomechanical properties of polymers plays an important role. Studies have shown that there is a functional relationship between the energy barrier of ethylene polymerization and the surface energy of MgCl2, which is consistent with the Sabatier principle [1].

The effect of the inclusion of cellulose nanocrystals on the mechanical properties of polyester resins as well as the development of polyester nanocomposites reinforced with continuous glass fiber, which combine traditional composites with additional advantages of nanocomposites, has been studied [2]. The addition of 4% cellulose nanocrystals to the polyester matrix leads to optimal tensile and fatigue properties. The mechanical properties were improved due to the improved structure of the material and the correct choice of compatible nanoparticles, as well as the addition of cellulose nanocrystals in the weight fraction, which did not adversely affect the mechanical properties of polyester nanocomposites reinforced with glass fiber.

The production processes and their influencing factors, mechanical characteristics, and recent applications of polyester reinforced with natural fiber and filled with nanoparticles have been considered [3]. Various methods of processing thermosetting materials reinforced with natural fibers have been presented, such as manual laying, molding with a vacuum infusion of resin, molding in matched dies, molding with resin transfer, thread winding, and pultrusion.

In [4], the effect of the inclusion of magnesium hydroxide as an antipyrine on the mechanical properties of high-density polyethylene, which include behavior under tension, bending, compression, and shear and fracture toughness, was investigated. The tensile yield strength steadily decreased when loading the filler to 10%, and then stabilized to 50%. The elongation at break sharply decreased between zero and 10% of the weight fraction, followed by a slight decrease to 50% of the mass fraction. The bending strength reaches a peak with an increase in the mass fraction of magnesium hydroxide. The strain rate had a significant impact on the tensile properties of the samples. The compressive strength and shear strength of the samples were directly proportional to Mg(OH)2 to 40% and 30%, respectively. Environmentally safe fire-resistant melamine polyphosphate (MPP) [5] was prepared and introduced into the composition of linear low-density polyethylene (LLDPE). In this study, the effect of this halogen-free flame retardant on the mechanical characteristics of LLDPE was studied. It was observed that Young's modulus and bending modulus increased almost linearly; by analogy, the tensile strength and tensile yield strength increased with an increase in the mass fraction of MPP. The elongation at break sharply decreased in the range from 0 to 10% by weight, followed by a slight decrease to 30% by weight. Flexural strength reached a peak with an increase in the mass fraction of MPP. The compressive and shear strength of the samples was directly proportional to the content of MFP up to 15% and 30%, respectively. The innovation in MPP content optimization, as well as the method of preparation, open up a wide range of applications.

The quantitative model of the RGO aspect ratio was proposed to study the relationship between the structure and performance characteristics of PPR/RGO composites [6]. The authors found that after the inclusion of RGO in poly-propylene with a concentration of 0.5 wt.% there was an improvement in mechanical properties (for example, tensile strength and toughness increased by 23% and 34%, respectively), and thermophysical properties, which include thermal stability (for example, the deposition temperature increased by 30 °C).

The process of electrophoretic deposition (EPD) of polyaniline (PANI) film was described kinetically using three models, namely, Hamaker, Zhang, and Baldisseri [7]. The kinetics of PANI-EPD were initiated as a function of the deposition stress and time, followed by microscopic characterization of the deposited films. The experimental results showed that it followed the linear law of sediment growth, which is consistent with the predictions of the law of Hammock. The presented study can provide an opportunity to evaluate the process of thermal degradation, which can also help to avoid the unintentional thermal degradation of polymer products in various industries.

The introduction of carbon fillers [8] can improve the thermal and electrical conductivity of polymer composites, but also has a significant effect on their bending and stretching behavior. Two types of carbon fillers were added to polypropylene: MWCNT and synthetic graphite. It was found that carbon fillers lead to a significant increase in the bending and stretching modulus of polypropylene composites. The maximum bending and tensile strength increased slightly with the addition of graphite, but it was significantly increased in the case of MWCNTs, since MWCNTs have exceptional rigidity and strength, and the ratio of their length to diameter is much larger compared to graphite.

Carbon nanostructures (CNSs) are used as modifying additives in various materials to improve their electro- and heat-conducting properties, mechanical strength, and thermal stability [9,10]. The successful employment of carbon nanotubes (CNTs) and graphene structures in composites with the ability to shield electromagnetic waves of a wide range was also reported [11]. Among the methods of combining carbon nanomaterials with polymer matrices of various natures, one of the most important places is occupied by

functionalization through the formation of surface groups of various natures that provide chemical, electrostatic, or van der Waals interaction with matrix macromolecules, thereby improving the uniformity of dispersion and noticeably increasing the effect of the introduction of a modifier [12,13]. The introduction of nanostructured carbon materials into polyurethane, as well as in the case of other polymers, contributes to a noticeable improvement in some properties. It was found that MWCNTs additives in polyurethane at a concentration of 2.5 wt.% ensure the efficiency of shielding electromagnetic interference at the level of 45 dB [14]. In addition, MWCNTs mixed with $Ti_3C_2Tx$ can improve the radio-absorbing properties of polyurethane and provide absorption up to 70 dB in the general X-band and Ka-band at a thickness of 200 μ [15]. Recently a functional radio-absorbing composite with the property of self-healing was obtained based on CNT and polyurethane [16]. The application of CNTs to increase the electrical conductivity of this polymer looks promising [17].

To implement carbon nanostructures into polyurethane the ultrasound treatment was successfully applied [18]. However, the difficulty of performing this in viscous media needs to be acknowledged, and with an increase in the concentration of the filler, it becomes ineffective.

There are multiple reports on various methods of functionalization of CNSs for adaptation to polyurethane. Chemical treatment of CNTs is often time-consuming and includes multiple stages (e.g., [19]). CNTs were pre-mixed with p-aminophenylpropargyl ether in an isoamyl nitrite medium, resulting in an alkylated form of CNT (the functional group contains a triple bond), which was then attached to the polymer chain during the catalytic reaction. For better dispersion of CNTs in polyurethane, modifying them with organosilanes was proposed, in particular, γ-aminopropyltrimethoxysilane [20]. This causes the grafting of groups with a terminal NH2 fragment, capable of forming covalent bonds with matrix macromolecules.

Often the functionalization of CNTs by organosilanes for introduction into polyurethane is a multi-stage process [21]. It includes:

(1) Oxidation of CNTs with concentrated nitric acid to form carboxyl groups;
(2) Reduction of carboxyl groups on the surface of CNTs to hydroxyl groups through interaction with $LiAlH_4$;
(3) Silanization of hydroxylated CNTs as a result of their interaction with [3-(2-aminoethyl)aminopropyl]trimethoxysilane in an aqueous-methanol medium, followed by washing from excess reagents and vacuum drying.

Then silanized CNTs are introduced into polyurethane during polymerization and growth of macromolecules.

Graphene materials (classical single-layer graphene, graphene oxide (GO), and graphene nanoplastics (GNP)) could also be a part of polyurethane composites. Due to the strong van der Waals interaction between graphene layers, exfoliation of individual graphene sheets in a polymer matrix is a very difficult task. At the same time, reduced GO and functionalized (for example, with hydroxyl or carboxyl groups) graphene has very good compatibility with polyurethane and other polymer matrices. Due to the ordered chemical bonds with the polymer matrix, it is possible to form correct oriented structures based on graphene layers in the volume of the composite [22]. Other ways of graphene functionalization are also possible. It should also be taken into account that the final properties of graphene/polyurethane composites depend on the method of introduction of the nanomodifier. Thus, in [23], oxidized graphene reduced by GO and treated with GO isocyanate in dimethylformamide according to the method [24] was used to modify thermoplastic polyurethane.

At the same time, various methods of combining the nanomodifier with the matrix were used: mixing of components in a solvent (dimethylformamide), in situ polymerization, and compounding of the melt (at 180 °C). It was shown that with the first method, a uniform distribution is not achieved, with the second, a relatively uneven dispersion is formed; with the third, a tightly stitched network is formed in the volume of the matrix. Functionalization by isocyanate and oxide groups, in turn, promotes the combination of graphene structures with the matrix. The authors themselves point out the disadvantages of the proven method

of obtaining a composite by mixing components in dimethylformamide since it requires the removal of excess solvent and is non-alkaline. Functionalization makes it possible to obtain composites by extrusion, which is the preferred method.

Another method of obtaining graphene/polyurethane composites by injection molding has proved effective when using hexadecylated graphene structures [25] obtained from CNTs as a result of their intercalation in Na/K melt with subsequent treatment with 1-iodohexadecane as a filler for thermoplastic polyurethane [26]. In [27], graphene oxide is functionalized by polyesters, as a result of which it acquires the ability to integrate into the polymer matrix. In [28], oxidized graphene was introduced into a mixture consisting of bifunctional polycaprolactone, 1,4-butanediol, 4,4-methylene bis (phenylisocyanate), a catalyst, and dimethylformamide, and kept under stirring for a day at 60 °C in a nitrogen atmosphere. The resulting dispersion was mixed with a polyurethane solution in DMFA, after which the composite was formed as a result of the prolonged evaporation of the solvent.

Amino groups on the surface of CNSs formed in various ways may also have an affinity for polyurethane macromolecules. It was shown that the treatment of oxidized graphene with 3-aminopropyltriethoxysilane in an aqueous alcohol medium with subsequent reduction with hydrazine hydrate promotes the formation of functional groups on their surface with NH2 fragments facing the outside [29]. A composite of graphene functionalized by this method with polyurethane was proposed by in situ polymerization.

According to [30], conductive polymers (polyaniline, polypyrrol, and polythiophene) are compatible with polyurethane. At the same time, the extrusion introduction of fillers of this type into the matrix is possible only at low concentrations and is rarely used. e.g., a method is presented that does not involve the solvents to obtain a composite of a mixed epoxy-polyurethane matrix modified with an emeraldine polyaniline base [31].

One paper [32] presented a method for the production of a polyurethane composite in which MWCNTs, as filler, were modified with polyaniline. Dried MWCNTs/polyaniline and polyurethane were mixed with tetrahydrofuran, then the solvent was removed by vacuum drying.

UV ozonation of the CNT's surface can be used as a simple functionalization method [33]. The composite based on thermoplastic polyurethane elastomer and surface-activated CNTs acquires improved mechanical and electrophysical properties and can be obtained by mixing in a melt.

In [34], the interesting technique of transferring the pattern from CNTs was reported, which was pre-printed by inkjet printing on polyethylene terephthalate (PET) film, and then transferred to PDMS, which made it possible to form CNT patterns. The diaphragm made of PDMS with CNT can act as a sensor. This is due to the pressure causing a deviation in the shape of the diaphragm, followed by a change in the electrical resistance of the CNT pattern [35].

The employment of the heat release effect during the flow of electric current in a polymer matrix with MWCNTs [36,37] allows the production of electric heaters with a high level of energy efficiency. For electric heaters, conductive networks in the structure of the dielectric matrix can be obtained using CNT [38], graphene [39], or SnO2 doped with fluorine (FTO) with metal nanotubes (Cr-nd, NiCr-nd, and Ni-nd) [40]. For epoxy resin modified with CNT, the physical and mechanical properties depend on the morphology of CNT, as well as the distribution features [41]. To obtain an electric heater, graphene and polyvinyl alcohol (PVS) were used [42]. The heating temperature for PVS with graphene reached 60 °C at a voltage of 10 V in 180 s. To obtain a flexible heater, a polyurethane matrix was used [43]. However, most of these flexible heaters did not stretch [44], and the polyurethane matrix [45] showed a decrease in the heating power during stretching due to the destruction of the conductive network.

It is necessary to take into account the possibilities of conductive polymer nanocomposites with reversible dynamic bonds, as well as their energy activation for self-healing through the Joule effect [46]. An alternative option for MWCNTs is to use graphene as a conductive

additive to improve the electrical and thermal conductivity of polymers [47]. The implementation of additional functional properties should be noted [48]. In [49], nanocomposites based on graphite oxide and natural rubber were developed using pre-mixing technology.

Electric heaters based on organosilicon elastomers have a functional feature associated with self-regulation of the heating temperature [50,51]. The elastomer deformations are the focus of studies [51]. The effect of irradiation of a composite with MWCNTs on thermomechanical and electrical properties and applications in electric heating materials was established [52]. To obtain thermoplastic polymer composites with improved thermal and mechanical properties, as well as durability, polyurethane-coated carbon fiber (CF) and electron beam irradiation (EB) were used as effective reinforcing fillers and improved crosslinking processes [53]. For this purpose, composites based on polyamide 6 (PA) with different HC content from 1 to 10 wt.% were made by compounding from melt and compression molding and then irradiated with various doses of EP from 50 to 200 kGy. SEM and FT-IR data show that CF is well dispersed in the PA matrix and has good interfacial adhesion due to the formed intermolecular interaction, which is enhanced for composites with crosslinked PA matrices after EB irradiation.

Flexible composites such as silicone rubber (SR)/carbon fiber (CF)@polydopamine (PDA) were reported [54]. CF was modified by oxidative self-polymerization of dopamine (DA), and the effect of DA concentration on the properties of SR/CF@PDA composites was studied. The research demonstrated that, compared to another surface modification process, carbon fiber modified with dopamine is cost-effective, "green", and highly effective. The advantage of SR/CF@PDA is excellent interphase coupling in the material structure and flexibility.

The goal of the current study is the analysis of the effect of MWCNTs concentration on heat release in nanomodified elastomers under tension and torsion during electric current flow. The detailed tasks are:

(1)　Manufacturing of elastic organosilicon matrices of modified MWCNTs synthesized by CVD technology;
(2)　The study of electrically conductive nanomodified elastomers of heat release under tension and torsion under the action of electric voltage.

## 2. Materials and Methods

The silicon-organic compound Silagerm 8030 (ELEMENT 14 LLC, Moscow, Russia) (Table 1) was employed as the polymer matrix of the elastomer. To obtain catalysts, we employed a multiple-stage approach which includes dissolution of the initial components (with and without ultrasonic treatment) and heating of the resulting solution (decomposition/calcination). Ultrasonic treatment of the solution of the initial components of the Ni/MgO catalyst increases efficiency.

**Table 1.** Characteristics of silicon-organic compound Silagerm 8030.

| # | Parameter | Value |
|---|-----------|-------|
| 1 | Shore A hardness | 25–35 |
| 2 | Compound lifetime, min, at 20 °C for at least | 30 |
| 3 | Relative elongation at break, %, not less | 450 |
| 4 | Tensile strength, MPa, not less | 3.5 |
| 5 | Component ratio (paste/hardener) | 1/1 |

For the treatment of the precatalyst solution with ultrasound in the I10 unit (with a frequency of 22 kHz and a power of 2 kW) (Inlab LLC, St. Petersburg, Russia), (volume 50 mL), a time delay of 10 s was selected, this activation time was sufficient to increase the efficiency of the formed catalyst by 20–30%. The dissolution temperature of the initial components should not exceed 60 °C, and the decomposition of the precatalyst (Ni/MgO) was carried out for 30 min at 500 °C. For use in the process of MWCNTs synthesis, the

resulting catalyst was pre-crushed. The obtained catalysts were used for the synthesis of a nanomodifier by gas-phase chemical deposition (CVD). The synthesis of MWCNTs using CVD technology was carried out in the installation shown in Figure 1.

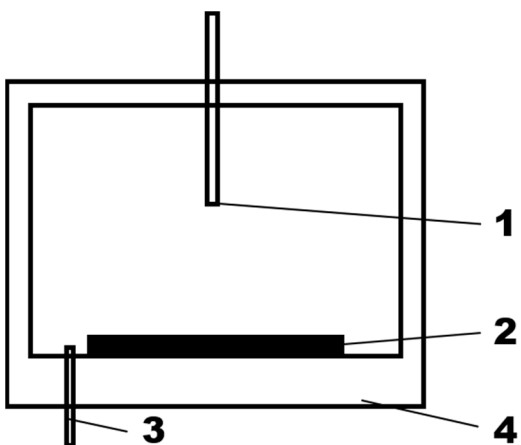

**Figure 1.** The schematic of the MWCNTs synthesis plant. 1—a device for a vapor–gas mixture; 2—heating device on which the catalyst is located; 3—the device for connecting a vacuum pump; 4—housing.

The yield of MWCNTs synthesized on the studied catalytic system ($\gamma$) was determined by the formula:

$$\Gamma = (m2 - m1)/(m1 - m0)$$

where m0 is the mass of the substrate, m1 is the mass of the substrate with the catalyst, and m2 is the mass of the substrate with the catalyst and the MWCNTs. We obtained in total three catalytic systems, based on components' variations.

The morphology of MWCNTs (SEM, TEM) was studied on a Hitachi H-800 microscope (Hitachi Ltd., Tokyo, Japan) with an accelerating voltage of up to 200 keV. Surface-mapping of nanomodified elastomers was carried out on a Raman Microscope DXR (Raman Microscope Termo Scientific) (Thermo Scientific Ltd., London, UK). The wavelength of the exciting laser was 532 nm.

To remove moisture from the MWCNTs before introduction into the elastomer, a drying cabinet Sanyo Convection Oven MOV 210F (Sanyo Ltd., Osaka, Japan) was employed at a temperature of 110 °C.

To obtain samples of nanomodified elastomers, the technology presented in [55] was applied. Component (A) was an organosilicon compound and MWCNTs mixed on a top-drive mechanical mixer WiseStir HT 120DX (WiseStir Ltd., Seoul, Korea) at 200 rpm for 20 min. In the next step, a second Pt-based component (Pt) was introduced into the mixture, providing polymerization (B), followed by stirring for 10 min at a temperature of 22 °C. The concentration of MWCNTs in the elastomer varied from 1 to 8 wt.% in increments of 1%. The length of the elastomer sample with MWCNTs is 10 cm, width 1 cm, and thickness 1.5 mm. At the same time, the length of the working area is 7 cm, since 1.5 cm from each edge forms a contact pad and fastening. The obtained products are listed in the Table 2.

**Table 2.** Elastomers modified with MWCNT.

| MWCNTs Content in Elastomer, % | Elastomer's Designation | | |
|---|---|---|---|
| | Ni/$_{0.16}$MgO | Ni/$_{0.3}$MgO | Ni/$_{0.5}$MgO |
| 1 | NCOC 1 (0.16) | NCOC 1 (0.3) | NCOC 1 (0.5) |
| 2 | NCOC 2 (0.16) | NCOC 2 (0.3) | NCOC 2 (0.5) |
| 3 | NCOC 3 (0.16) | NCOC 3 (0.3) | NCOC 3 (0.5) |
| 4 | NCOC 4 (0.16) | NCOC 4 (0.3) | NCOC 4 (0.5) |
| 5 | NCOC 5 (0.16) | NCOC 5 (0.3) | NCOC 5 (0.5) |
| 6 | NCOC 6 (0.16) | NCOC 6 (0.3) | NCOC 6 (0.5) |
| 7 | NCOC 7 (0.16) | NCOC 7 (0.3) | NCOC 7 (0.5) |
| 8 | NCOC 8 (0.16) | NCOC 8 (0.3) | NCOC 8 (0.5) |

For the study of elastomer samples under tension and torsion, clamps made of PLA polymer on a Zenit Duo 3D printer (3Dtool Ltd., Moscow, Russia) were used. The clips were placed in the tripod mounts (Figure 2a–d).

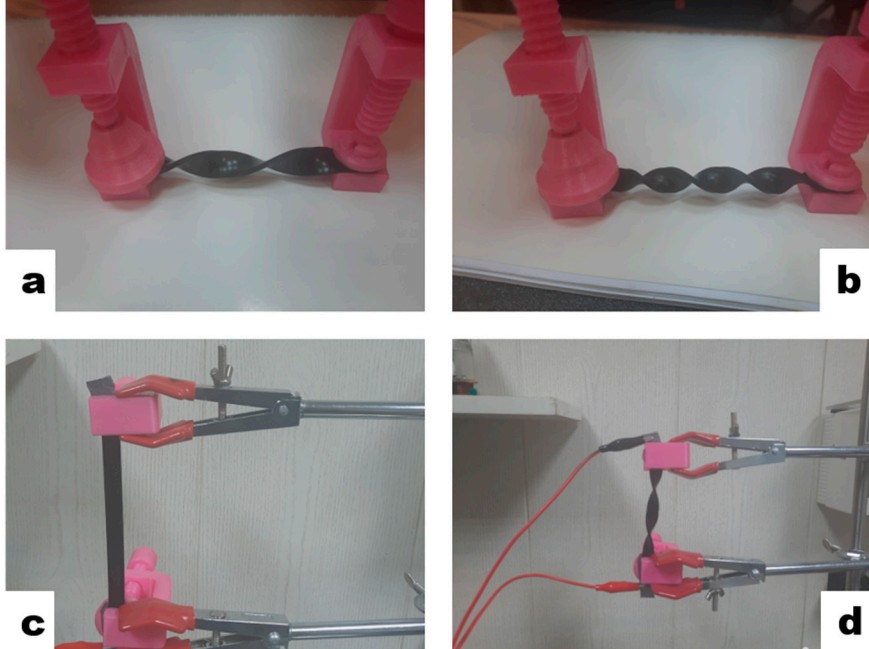

**Figure 2.** The samples under different tensions. (**a**)—360° torsion, (**b**)—720° torsion, (**c**)—tension, and (**d**)—360° torsion with connected electrodes.

Resanta LATR TDGC2-3″ (Resanta Ltd., Moscow, Russia) was used as a power source, with a control range from 0 to 260 V. The measurement of specific volumetric electrical conductivity was carried out using a "Teraohmmeter E6-13a" (Punane-Ret Ltd., Tallinn, Estonia) with a range of measuring electrical resistance up to 14 vol. The elastomer samples were stretched and twisted. At the same time, when stretched, the working area of the elastomer changed from 7 cm to 8.5 and 9.5 cm.

To study the temperature field, a non-contact measurement method was used using a thermal imager Testo-875-1 with an optical lens 32 × 23° (SE & Co. KGaA, Testo, Lenzkirch, Germany) with a distance of 10 cm from samples of nanomodified elastomers in a darkened room without access to sunlight. The temperature of nanomodified elastomers was measured by a two-channel thermometer Testo 992 (SE & Co. KGaA, Testo, Lenzkirch, Germany), the surface temperature was determined, and based on the data obtained, a comparison was made with the temperature recorded by the thermal imager, after which

the radiation coefficient used for further measurements was selected. The obtained thermal images of elastomers were analyzed using IRSoft v4.9SP1 software (SE & Co. KGaA, Testo, Lenzkirch, Germany). To conduct TG and DSC studies, the simultaneous thermal analyzer NETZSCH STA 449F3 (NETZSCH-Gruppe, Selb, Germany) was used. The tests were carried out in an argon atmosphere.

## 3. Results

For the synthesis of MWCNTs, three Ni/MgO catalytic systems were used, differing in the content of the active component: Ni/0.16MgO; Ni/0.3MgO; and Ni/0.5MgO. All Ni/Mo catalysts containing more than 50% of the active component had a coral-like structure (Figure 3a). With a decrease in content in the catalyst composition, their structure changed (Figure 3b).

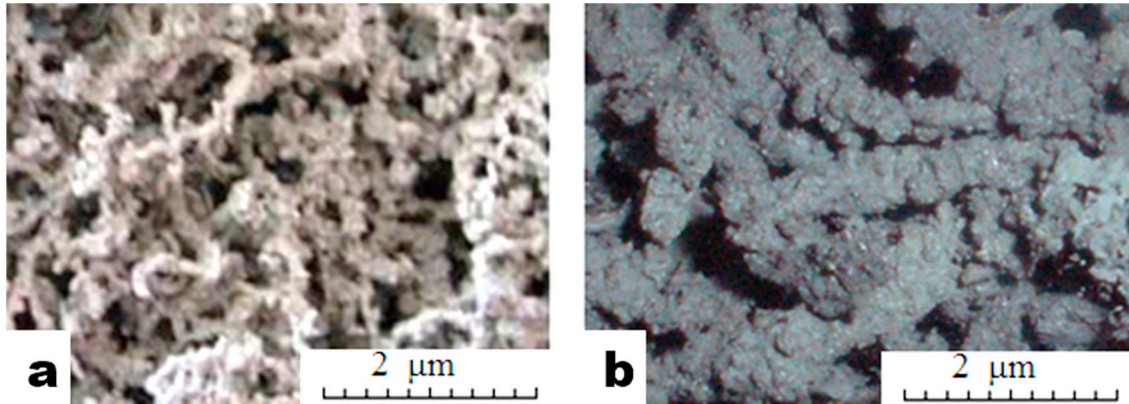

**Figure 3.** The structure of Ni/MGO catalysts. (**a**)—More and (**b**)—less than half of the active component.

These catalysts were used for the synthesis of MWCNTs by the CVD method. The characteristics of the obtained catalysts are presented in Table 3.

**Table 3.** Characteristics of Ni/MgO catalysts.

| Composition | Specific Surface Area, $m^2/g$ | Effectiveness, $g_C/g_{cat}$ |
|:---:|:---:|:---:|
| 9:1 | 51.9 | 9.64 |
| 8:2 | 55.6 | 11.30 |
| 7:3 | 60.8 | 4.90 |

The synthesized nanomaterial was studied using scanning (SEM) and transmission (TEM) electron microscopy. The morphology and structure of MWCNTs synthesized on the obtained catalysts are shown in Figure 4.

The analysis of SEM images allows us to assess the morphological properties of MWCNTs samples, which are based on filamentous structures. The synthesized MWCNTs on Ni/$_{0.3}$MgO and Ni/$_{0.16}$MgO catalysts are characterized by clearer morphological contours. The diameter of carbon filamentous formations synthesized on Ni/$_{0.16}$MgO and Ni/$_{0.3}$MgO catalysts have a diameter of ~30–60 nm. In the sample obtained on a Ni/$_{0.5}$MgO catalytic system, in addition to filamentous formations, there were a lot of unreacted catalysts.

Analysis of the structure of the obtained MWCNTs allows us to conclude that these are MWCNTs ~40 nm containing nickel metal particles at their ends. The defect of the synthesized nanomaterial was determined by Raman spectroscopy. The value of the peak intensity ratio D/G makes it possible to determine the degree of defectiveness of graphene layers of synthesized carbon nanostructures.

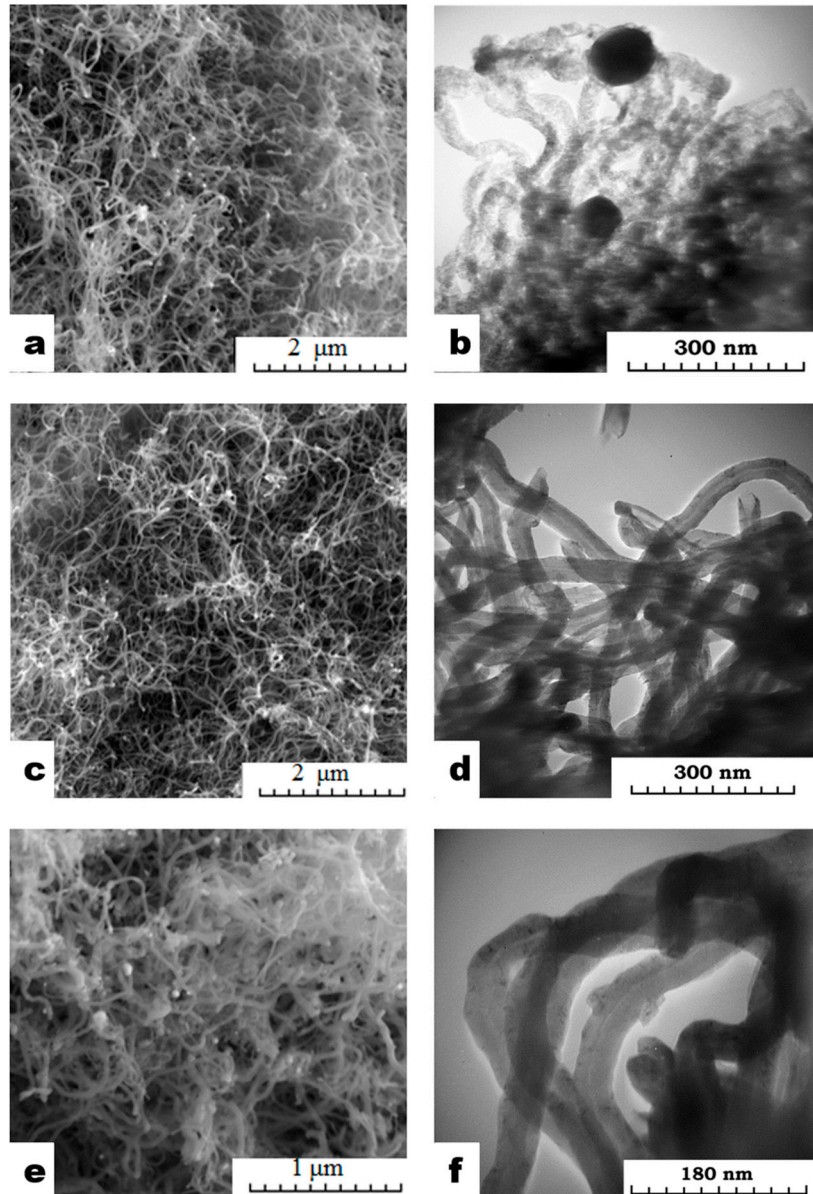

**Figure 4.** Morphology and structure of MWCNTs synthesized on (**a,b**)—Ni/$_{0.3}$MgO, (**c,d**)—Ni/$_{0.5}$MgO, and (**e,f**)—Ni/$_{0.16}$MgO catalysts.

The Raman spectra of MWCNTs synthesized on the obtained Ni/MgO catalyst samples are shown in Figure 5.

According to Figure 5, the degree of defectiveness of graphene layers of MWCNTs (D/G) synthesized on a Ni/$_{0.3}$MgO catalyst was 1.44, on Ni/$_{0.5}$MgO—1.1 and on Ni/$_{0.16}$MgO—1.64. A comparative analysis of the results of Raman mapping of the surface of nanomodified elastomers with a catalyst for the synthesis of MWCNTs Ni/$_{0.5}$MgO is presented in Figure 6a–f for MWCNTs concentrations from 3 to 8%.

The Raman mapping of the surfaces of elastomers with MWCNTs shows the presence of a combination of both the dielectric phase (associated with the elastomer) and the conductive phase (associated with MWCNTs). An increase in the uniformity of the MWCNTs distribution in the elastomer structure was observed at mass concentrations of 7 wt.% NCOC 7(0.5) (Figure 6e) and 8 wt.% NCOC 8(0.5) (Figure 6f). A change in the mass concentration of MWCNTs, starting from a value of 7%, in the elastomer by 1% led to an increase in electrical conductivity from 1.8 (NCOC 7(0.5)) to 2 (NCOC 8(0.5)) S/cm. The

micro-dimensional agglomeration characteristic of MWCNTs [46] ensures the formation of electrically conductive contacts.

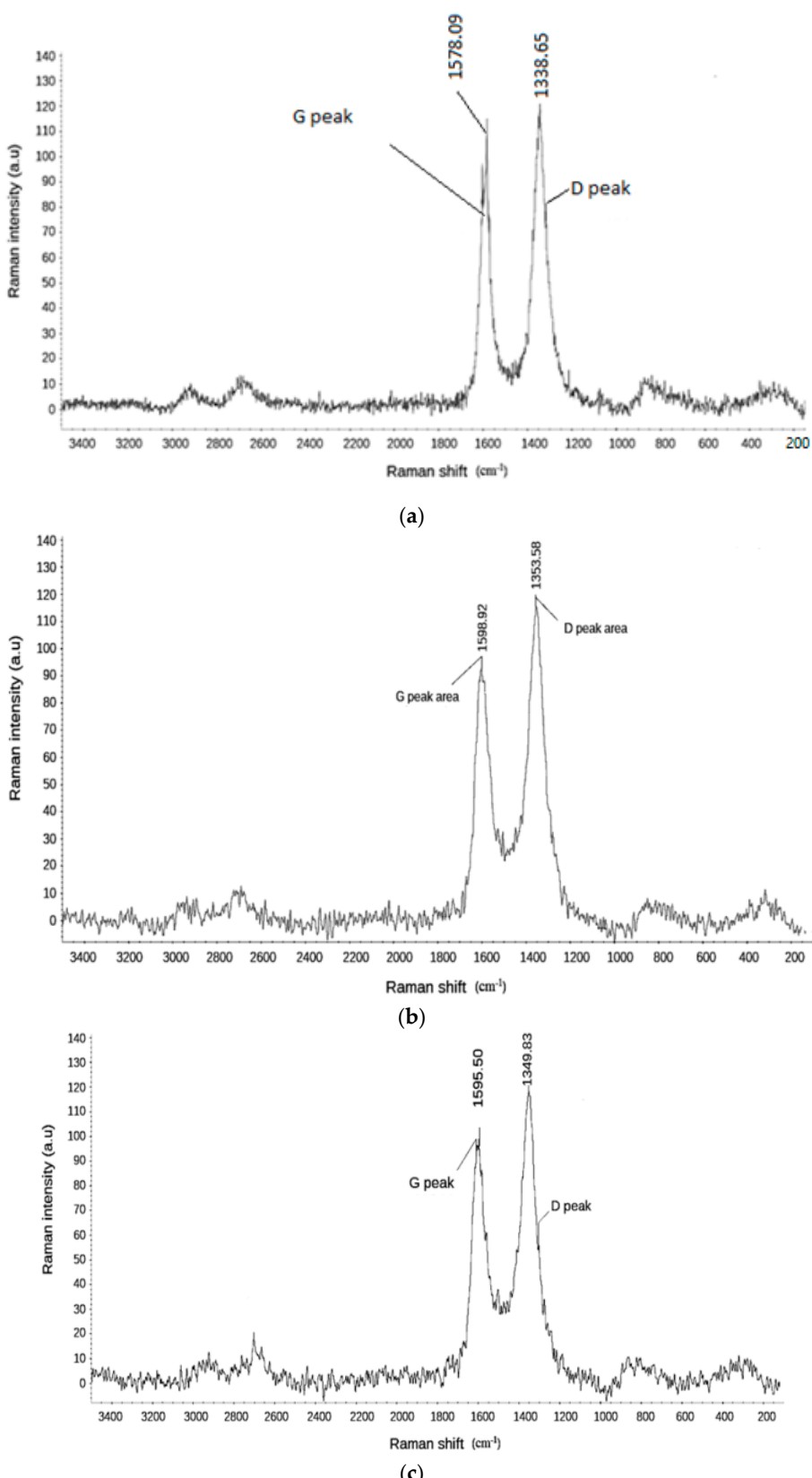

**Figure 5.** Raman spectrum of MWCNTs synthesized on (**a**)—Ni/0.3MgO, (**b**)—Ni/0.5MgO, and (**c**)—Ni/0.16MgO catalysts.

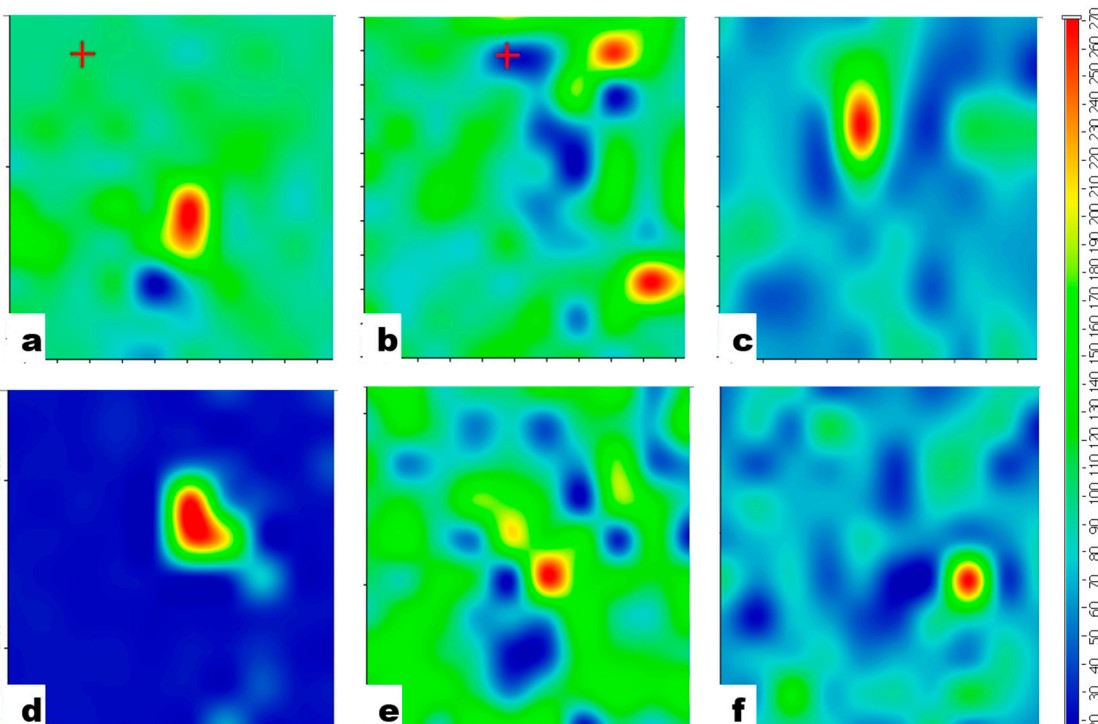

**Figure 6.** Raman mapping of the surface of nanomodified elastomers: (**a**)—3%; (**b**)—4%; (**c**)—5%; (**d**)—6%; (**e**)—7%; (**f**)—8%.

In the case of NCOC X(0.3)), electrical conductivity started at 1.6 for 7% MWCTNs (NCOC 7(0.3)) and reached electrical conductivity of 1.9 S/cm for 8% MWCNTs (NCOC 8(0.3)). Further decrement of Ni content (NCOC X(0.16)) caused growth of electrical conductivity from the value of 1.4 for 7% MWCNTs (NCOC 7(0.16)) to 1.7 S/cm for 8% MWCNTs (NCOC 8(0.16)).

Samples with concentrations of MWCNTs from 1 to 5% had low electrical conductivity (NCOC 1—$2.8 \times 10^{-8}$ S/cm; NCOC 2—$4.5 \times 10^{-7}$ S/cm; NCOC 3—$3.7 \times 10^{-6}$ S/cm; NCOC 4 $5 \times 10^{-5}$ S/cm; NCOC 5—0.08 S/cm) and in the voltage range from 0 to 260 V at ambient temperature were not heated. The same is typical for a series of samples NCOC 1–8 (0.3–0.16). At a mass concentration of MWCNTs in an elastomer equal to 6 wt.%, it had a resistance of 0.9 S/cm and a voltage equal to 20 V when it was heated.

The thermograms shown in Figure 7a–c were obtained for elastomers with a mass concentration of MWCNT: (Figure 7a) 6, (Figure 7b) 7, and (Figure 7c) 8%. An increase in the concentration of MWCNT at a comparable supply voltage causes an increase in temperature from 57.4 at 6 wt.% to 74.6 °C at 8 wt.%.

Panels d and e of Figure 7 demonstrate thermograms of the elastomer in its initial state (maximum temperature 60.5 °C) and stretched by 20%. This study was conducted for an elastomer with MWCNTs (8 wt%) with a supply voltage of 50 V. When stretching an elastomer sample with MWCNTs by 20% of the initial length (Figure 7e), a decrease in the maximum temperature by 8.2 °C (52.3 °C) was observed.

When the elastomer sample was torsioned at 360 °C (Figure 7f), sections of the elastomer with an elevated temperature were formed in the central zone (76.3 °C). When torsion at an angle of 540° (Figure 7e) over a period of 20 s, the temperature of the sample increased to 101.7 °C.

When stretching samples with a size of 7.5 cm, the electrical resistance increased from 3.5 to 4.15 kOhm with an elongation of 1.5 cm and a further increase in elongation by 1 cm increases the resistance to 4.22 kOhm (Figure 7a–c).

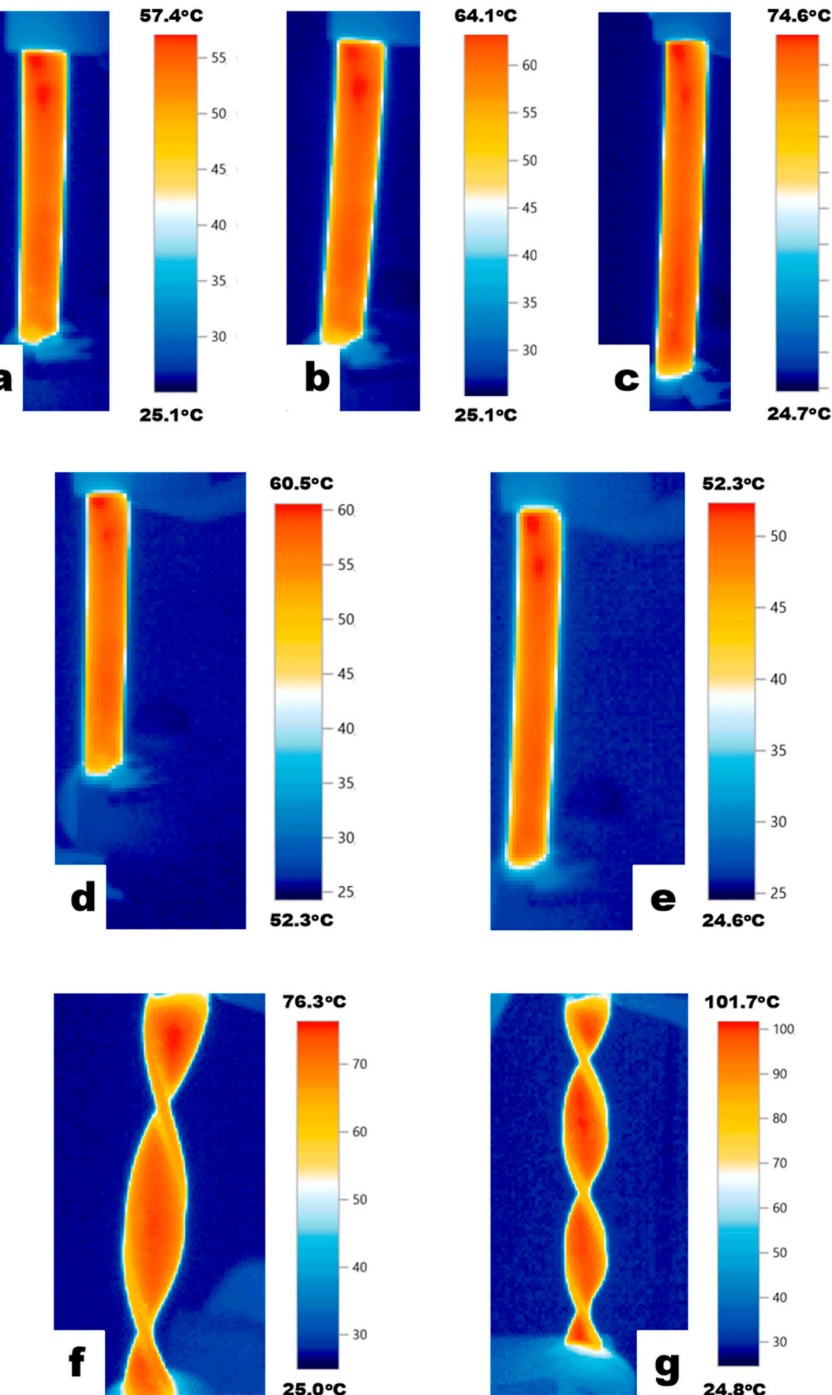

**Figure 7.** Thermograms of the surface of nanomodified elastomers. (**a–c**)—Thermograms of the surface for elastomers with a mass concentration of MNT: (**a**)—6, (**b**)—7, and (**c**)—8 w%. (**d,e**)—Thermograms of the surface of nanomodified elastomers: d is the initial sample; e is the sample after stretching by 20%. (**f**)—Thermogram of the surface of the nanomodified elastomer at 360° torsion. (**g**)—Thermogram of the surface of the nanomodified elastomer at 540° torsion.

Differences in the distribution of the temperature field on the surface of the investigated elastomer with MWCNTs during tension and torsion are associated with changes in the local electrical resistance in the areas subjected to mechanical deformations, which, on the one hand, is caused by geometric distortion of the dielectric matrix, and, on the other hand, by deformation of MWCNTs. At the same time, the stretching of the elastomer

causes a decrease in the temperature of heat release since there is an internal displacement in the conductive grid, formed by both individual MWCNTs and their agglomerates.

From the analysis of the influence of the concentration of MWCNTs on the heat release process, it follows that there is a percolation effect. The relationship of the electrical conductivity of the elastomer with the additives of MWCNTs can be expressed in the following way [56]:

$$\sigma = \sigma_c + (\sigma_m - \sigma_c)[(\varphi - \varphi_c)/(F - \varphi_c)]^t$$

where $\sigma_m$—electrical conductivity of the elastomer at the maximum mass content of conductive additives (S/cm),

$\sigma$—electrical conductivity of the elastomer (S/cm),

$\sigma_c$—electrical conductivity of the elastomer at the percolation threshold (S/cm).

$F$—maximum volume fraction of the filler (MWCNTs), %.

$\phi$—volume fraction of conductive filler, %.

$\phi_c$—volume fraction of conductive filler at the percolation threshold, %.

$t$—(degree of equation) or critical indicator.

During stretching and torsion of the elastomer, there is a change in parameters such as $F$ and $\sigma_c$, which causes a change in electrical conductivity and the subsequent heat release process.

Calculation of the electrical conductivity of the elastomer taking into account the tunnel effect:

$$R_{int}(d_c) = \frac{d_c \times \hbar^2}{Se^2\sqrt{2m\lambda}}exp\left(4\pi d_c/\hbar\sqrt{2m\lambda}\right)$$

where $d_c$ is the electric tunnel distance, m; $S$ is the contact area of the MWCTN, m²; $\lambda$—the height of the potential barrier of the polymer; $m$ is the mass of the electron, kg; $e$ is the electric charge of the electron, Kl; and $\hbar$ is Dirac's constant (Planck's constant divided by $2\pi$), m²kg/s. Namely, the changes in $S$ allow us to explain the peculiarities of the change in the electric heating process during tension and torsion.

Figure 8 demonstrates the dependence of electric conductivity from MWCNTs content in elastomer, % (6, 7, 8 wt.%). An increase in the mass concentration of the conductive filler in the elastomer matrix leads to a linear increase in conductivity in the concentration range from 2 to 7 wt.%, which is typical for all 3 types of MWCNTs obtained on the previously described catalysts. Increasing the concentration to 8 wt.% demonstrates the formation of a saturation line in which a further increase in the conductive filler does not lead to significant changes in the conductivity of the nanomodified composite.

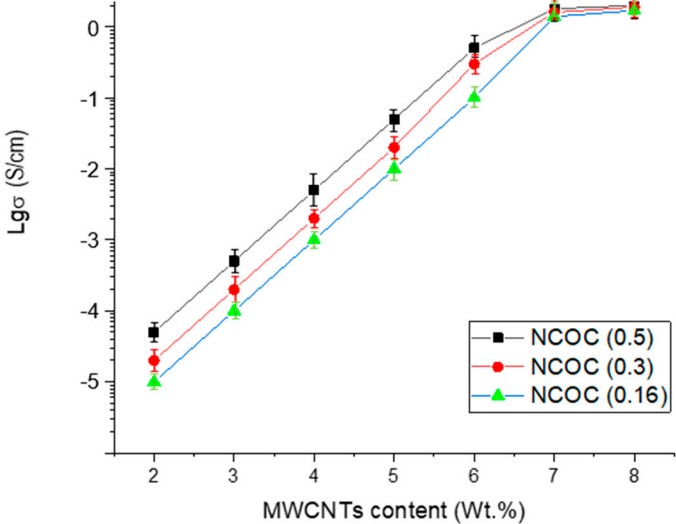

**Figure 8.** The dependence of electrical conductivity from MWCNTs content in elastomer.

The change in the heat release power *P(T)*₇ can be described by the temperature dependence for samples with a mass concentration of MWCNTs equal to 7 and 8% as

$$P(T)_7 = 7 - 20 \times 10^{-2}T + 5 \times 10^{-3}T^2 - 3 \times 10^{-5}T^3$$

and

$$P(T)_8 = 8.5 - 20 \times 10^{-2}T + 7 \times 10^{-3}T^2 - 2 \times 10^{-5}T^3$$

respectively.

The application of a nano-modified elastomer requires an analysis of the temperature processes occurring during its heating (Figure 9).

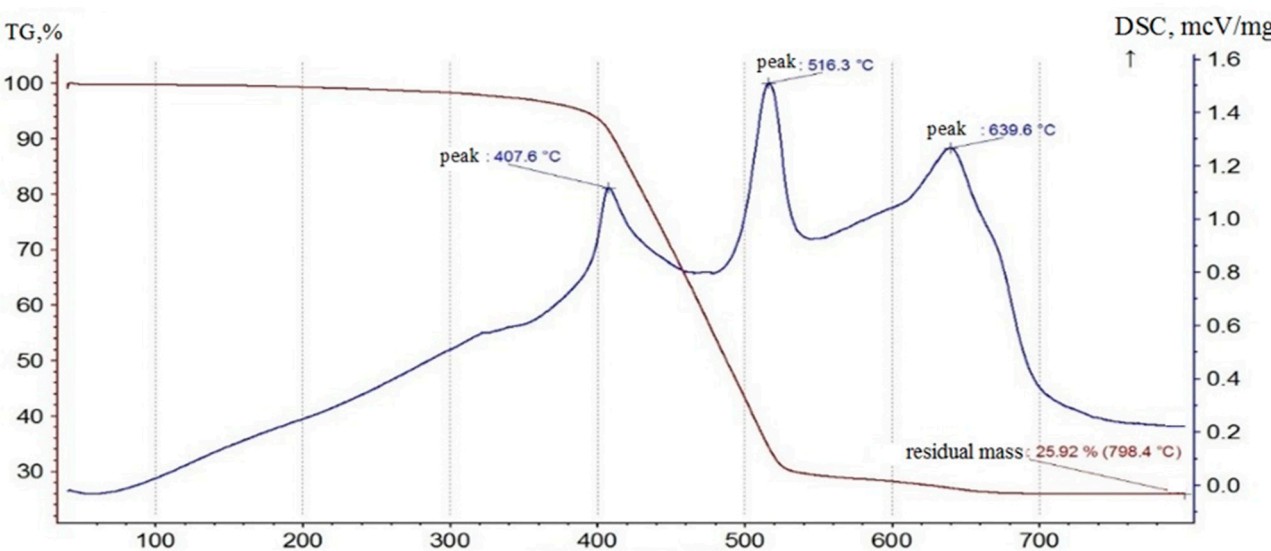

**Figure 9.** DSC of nano-modified elastomer. The heating process of the nanomodified elastomer clearly proceeds without any changes at temperatures up to 200 °C. Further heating up to 400 °C is accompanied by a phase transition.

The employment of a flexible heater makes it possible to create efficient electric heating systems that can be used in ventilation equipment, where in order to increase the efficiency of heat exchange, it is necessary to optimize the location of heaters in accordance with the movement of air flows. We need to acknowledge the possibility of using the obtained elastomer samples with MWCNTs as sensitive elements of strain sensors, which will allow obtaining information about physical and chemical parameters in accordance with the principles of measuring changes in electrical resistance that occurs during tension, compression and torsion.

## 4. Discussion

Heat release in elastomers with MWCNTs concentration from 1 to 8 wt % were studied when an alternating electric power in the range up to 250 V was applied. Samples with a concentration from 1 to 5 wt.% in the supply voltage range from 0 to 260 V were not heated when an electric current was flowing.

Samples with concentrations from 1 to 5% had low electrical conductivity (NCOC 1—2.8 × 10⁻⁸ S/cm; NCOC 2—4.5 × 10⁻⁷ S/cm; NCOC 3—3.7 × 10⁻⁶ S/cm; NCOC 4 5 × 10⁻⁵ S/cm; NCOC 5—0.08 S/cm) and in the voltage range from 0 to 260 V at ambient temperature were not heated. The same is typical for a series of samples NCOC 1–8 (0.3–0.16). At a mass concentration of MWCNTs in the elastomer equal to 6 wt.%, the sample had a resistance of 0.9 S/cm and a voltage equal to 20 V was heated.

When the elastomer sample was twisted with a 360° MNT, sections of the elastomer with an elevated temperature were formed in the central zone (76.3 °C). When torsion was at 540° with a period of 20 s, the temperature of the sample increased to 101.7 °C. This is

due to the fact that compression occurs in the elastomer structure and the formation of improved contacts at the points of contact of the CNT. At the same time, during heating, the heating temperature decreases to 52.3 °C, which is due to the reverse effect caused by an increase in the distance between the conducting particles of the CNT in the elastomer structure. The fixed maximum temperature of 101.7 °C may be optimal for a nanomodified elastomer, since the limit value corresponding to 200 °C was not exceeded. The DSC of the nanomodified elastomer clearly proceeds without any changes at temperatures up to 200 °C.

According to Figure 7, when stretching samples with a size of 7.5 cm, the electrical resistance increases from 3.5 to 4.15 kOhm with an elongation of 1.5 cv and a further increase in length by 1 cm increases the resistance to 4.22 kOhm. Elastomer samples have flexibility, elasticity, and high efficiency of electrothermal transformation. The resistance of the elastomer/MWCNTs changes at 20% tension.

The comprehensive studies carried out with an assessment of the effect of stretching and twisting of images made of elastic matrices with conductive nanoscale additives on heat release and changes in electrical resistance allow us to clarify important issues of practical application of multifunctional materials for strain gauges and electric heating systems.

## 5. Conclusions

Let us summarize the reported results:

(1)    The synthesis of CNT based on Ni/Mo (Ni/0.3MgO, Ni/0.5MgO, Ni/0.16MgO) was carried out. The obtained CNT were characterized using SEM and PEM, as well as Raman spectroscopy. SEM image analysis allowed us to evaluate the morphological properties of MWCNTs samples based on filamentous structures.

(2)    The electrical conductivity of samples of elastomers with CNT was studied. Optimal values of the CNT concentration in elastomers were revealed. The influence of stretching and torsion on the process of heat release in elastomers has been studied. The power of heat release is presented in the form of regression equations, where there is a dependence of heat release on temperature, applied tension strain, and electric heating.

**Author Contributions:** Conceptualization, A.V.S. and A.A.V.; methodology, A.V.S.; software, A.V.S. and A.A.V.; validation, A.V.S.; formal analysis, A.V.S.; investigation, A.V.S.; resources, A.V.S. and A.A.V.; data curation, A.V.S.; writing—original draft preparation, A.V.S., A.A.V., M.N., Y.M.S., E.P.D. and D.T.R.; writing—review and editing, A.A.V. and M.N.; visualization, A.V.S. and A.A.V.; Supervision, Y.M.S. and A.A.V.; project administration, E.P.D. and D.T.R.; funding acquisition, A.A.V. and A.V.S. All authors have read and agreed to the published version of the manuscript.

**Funding:** This paper has been supported by the RUDN University Strategic Academic Leadership Program (recipient A.A.V.)

**Institutional Review Board Statement:** Not applicable.

**Informed Consent Statement:** Not applicable.

**Data Availability Statement:** The data presented in this study are available on request from the corresponding author.

**Acknowledgments:** Alexandre A. Vetcher (IBTN (RUDN)) gratefully acknowledges that this paper has been supported by the RUDN University Strategic Academic Leadership Program (for A.A.V.).

**Conflicts of Interest:** The authors declare no conflict of interest. The funders had no role in the design of the study; in the collection, analysis, or interpretation of data; in the writing of the manuscript, or in the decision to publish the results.

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
