# Peer review of "The Effect of Multi-Walled Carbon Nanotubes on the Heat-Release Properties of Elastic Nanocomposites"

_jcs, doi:10.3390/jcs6110333_

Round 1

Reviewer 1 Report

The paper presents an interesting approach based on the The effect of multi-walled carbon nanotubes on the properties of heat release of elastic nanocomposites. However, the innovation of the current research work should be further highlighted and emphasized. At the same time, the authors should consider the following comments to greatly improve the quality of the paper.

1. In the abstract, add a final statement that highlights the importance of this research and its possible potentials. Also, introduce the problem in the initial lines of the abstract.

2. The introduction needs to be improved by relating to the mechanics of the studied materials and their mechanical characteristics. The references to be included are: 10.1177/0021998318790093, 10.1016/j.polymertesting.2017.09.009, 10.1016/j.compstruct.2021.114698, 10.1177/0731684417727143, 10.1002/app.46770, 10.1016/j.porgcoat.2022.107015.

3. Kindly add a table that describes the main physical and chemical properties of the raw materials used in this study.

4. Were the preparation methods described by the authors come in accordance with a certain standard or do they follow previous procedures?

5. Regarding to the morphology of MWCNTs (SEM, TEM), what were the operating parameters and magnifications used? Kindly add that information in the characterization section.

6. In Figure 4, usually we use same magnification number to compare between microstructures. Why did the authors use a different magnification scale in figure 4.f than figure 4.b and figure 4.d?

7. The discussion section is very short and it doesn't link between the lengthy characterizations performed. This section needs to be reworked.

8. The conclusion is missing. It has to summarize the research outcomes in short statements with clear observations.

Author Response

2022-10-10

To Reviewer 1

Dear Reviewer:

Thank you so much for your time and efforts to improve our submission to readers. As about your comments – let me respond to them in the order from your review:

  1. In the abstract, add a final statement that highlights the importance of this research and its possible potentials. Also, introduce the problem in the initial lines of the abstract.

We edited the abstract accordingly

  1. The introduction needs to be improved by relating to the mechanics of the studied materials and their mechanical characteristics. The references to be included are: 10.1177/0021998318790093, 10.1016/j.polymertesting.2017.09.009, 10.1016/j.compstruct.2021.114698, 10.1177/0731684417727143, 10.1002/app.46770, 10.1016/j.porgcoat.2022.107015

We add them to Introduction.

  1. Kindly add a table that describes the main physical and chemical properties of the raw materials used in this study.

The Table is added

  1. Were the preparation methods described by the authors come in accordance with a certain standard or do they follow previous procedures?

We added doi.org/10.18323/2782-4039-2022-2-121-132, according to which we operated

  1. Regarding to the morphology of MWCNTs (SEM, TEM), what were the operating parameters and magnifications used? Kindly add that information in the characterization section.

We added necessary information

  1. In Figure 4, usually we use same magnification number to compare between microstructures. Why did the authors use a different magnification scale in figure 4.f than figure 4.b and figure 4.d?

The different scale is useful for readers to better visually evaluate the morphology

  1. The discussion section is very short and it doesn't link between the lengthy characterizations performed. This section needs to be reworked.

The Discussion is edited accordingly

  1. The conclusion is missing. It has to summarize the research outcomes in short statements with clear observations.

The Conclusion is formulated accordingly

Sincerely

Dr. Alex Vetcher

Reviewer 2 Report

This manuscript presents a method in order to study the modes of heat release of nanomodified elastomers at a voltage of 50 V at different levels of tension. The authors discuss the data in detail and the conclusion is reliable. Furthermore, the work may be useful in some special area. I suggest the manuscript for publication after proper revision.

Some comments may be useful for the authors.

1. I suggest the authors carefully read and improve the readability of this paper alongside the language.

2. In the introduction part, the authors should show the advantage, disadvantages, and weaknesses of the studied works.

3. What’s the meaning of 0.16, 0.3, 0.5 in the bracket in Table 1.

4. The conclusion section also needs revisions. It should briefly describe the findings of the study and some more directions for further research. The authors should describe academic implications, major findings, shortcomings, and directions for future research in the conclusion section.

Author Response

2022-10-10

To Reviewer 2

Dear Reviewer:

Thank you so much for your time and efforts to improve our submission to readers. As about your comments – let me respond to them in the order from your review:

  1. I suggest the authors carefully read and improve the readability of this paper alongside the language.

We edited the submission accordingly

  1. In the introduction part, the authors should show the advantage, disadvantages, and weaknesses of the studied works.

We edited Introduction accordingly

  1. What’s the meaning of 0.16, 0.3, 0.5 in the bracket in Table 1.

We reformulated the related section accordingly

  1. The conclusion section also needs revisions. It should briefly describe the findings of the study and some more directions for further research. The authors should describe academic implications, major findings, shortcomings, and directions for future research in the conclusion section.

We formulated Conclusions section accordingly

Sincerely

Dr. Alex Vetcher

Reviewer 3 Report

It was an interesting topic to study the heating performance of WMCNT elastomer composites under different conditions. But the current manuscript was poorly written and presented. Suggest to reject the current version and it is necessary for the authors to re-write/re-organize the contents and experiments.

1. As the heating element, the changes of electrical resistance under different mechanical conditions are critical. However, this part had not been properly investigated.

2. The manuscript had a very long Introduction. However, the contents were very fragmentary, and the logic of the research background was very confusing. Most of the contents did not directly support the knowledge gaps and research motivations of the work. This section was expected to be rewritten in a more concise and logical way.

3. In Materials and Methods, the synthesis and yield of MWCNTs needed to be minimized, as this was not the core focus of the work. While the procedures and details for the preparation of the nanocomposite were required, making the work repeatable. For the heating testing, no duration control of the heating was mentioned, making the results not comparable.

4. Fig. 1, Fig. 3 and Tab. 2 did not directly support the research aim, which needed to be removed.

5. "S/m" was used as the electrical resistance, causing more confusion for the results and discussion.

6. Line 347-355: The defined variables did not match the equation.

7. Section 4 Discussion was very poor and stopped without critical analysis.

8. No conclusion was presented.

Overall, the work was poorly presented, and it has to be rejected.

Author Response

2022-10-10

To Reviewer 3

Dear Reviewer:

Thank you so much for your time and efforts to improve our submission to readers. As about your comments – let me respond to them in the order from your review:

  1. As the heating element, the changes of electrical resistance under different mechanical conditions are critical. However, this part had not been properly investigated.

We edited the related section accordingly

  1. The manuscript had a very long Introduction. However, the contents were very fragmentary, and the logic of the research background was very confusing. Most of the contents did not directly support the knowledge gaps and research motivations of the work. This section was expected to be rewritten in a more concise and logical way.

We edited Introduction accordingly

  1. In Materials and Methods, the synthesis and yield of MWCNTs needed to be minimized, as this was not the core focus of the work. While the procedures and details for the preparation of the nanocomposite were required, making the work repeatable. For the heating testing, no duration control of the heating was mentioned, making the results not comparable.

We edited Materials and Methods, but left the description of synthesis method to let readers evaluate the history of material

  1. 1, Fig. 3 and Tab. 2 did not directly support the research aim, which needed to be removed.

We believe that this material is useful to readers for evaluation the material.

  1. "S/m" was used as the electrical resistance, causing more confusion for the results and discussion..

We do not use S/m at all and S/cm we use to characterize conductivity. Resistance we express in kOhms

  1. Line 347-355: The defined variables did not match the equation.

We corrected this accordingly

  1. Section 4 Discussion was very poor and stopped without critical analysis.

We corrected Discussion accordingly

  1. No conclusion was presented.

The Conclusion is formulated accordingly

Sincerely

Dr. Alex Vetcher

Reviewer 4 Report

The work reported in this manuscript is interesting and well presented. However, there should be further improvement and revision before the acceptance. Some comments are:

1.     Figure 3 and Figure 5 are fuzziness.

2.     Thermograms of the surface of nanomodified elastomers after stretching or torsion showed the temperature of the sample increases. Why?

3. DCS should be supplemented for the nanocomposites.

4. Some articles must be cited in the introduction;Applied Surface Science, 2022, 589,153002.;Composites Part B: Engineering, 2022, 239, 109970.; Adv.

Compos. Hybrid. Mater 2022,5:104–12.

Author Response

2022-10-10

To Reviewer 4

Dear Reviewer:

Thank you so much for your time and efforts to improve our submission to readers. As about your comments – let me respond to them in the order from your review:

  1. Figure 3 and Figure 5 are fuzziness

We corrected Figures accordingly

  1. Thermograms of the surface of nanomodified elastomers after stretching or torsion showed the temperature of the sample increases. Why?

We edited related section accordingly

  1. DCS should be supplemented for the nanocomposites.

We added DCS data

  1. Some articles must be cited in the introduction;Applied Surface Science, 2022, 589,;Composites Part B: Engineering, 2022, 239, 109970.; Adv.

We added these references

Sincerely

Dr. Alex Vetcher

Round 2

Reviewer 1 Report

The authors have successfully improved the manuscript according to the given suggestions. It can be accepted for publishing.

Reviewer 2 Report

The manuscript can be accepted now.

Reviewer 3 Report

There are some improvements from the first version. But the readability of this manuscript is still very poor, especially for the Introduction section. The comments for the first version is still applicable to the Introduction of this version "Most of the contents did not directly support the knowledge gaps and research motivations of the work. This section was expected to be rewritten in a more concise and logical way." Suggest to reduce 50% of the length in Introduction and only to include directly related background information to clarify the knowledge gaps and the new contribution of the current work.

Reviewer 4 Report

Acceptable